# Initial Contact with Forefoot or Rearfoot in Spastic Patients After Stroke—Three-Dimensional Gait Analysis

**DOI:** 10.3390/neurolint17010010

**Published:** 2025-01-18

**Authors:** Inês Mendes-Andrade, Miguel Reis e Silva, Jorge Jacinto

**Affiliations:** Laboratório de Marcha, Centro de Medicina de Reabilitação de Alcoitão, 2649-506 Alcabideche, Portugal; miguelreisesilva@icloud.com (M.R.e.S.); jor.jacinto@netcabo.pt (J.J.)

**Keywords:** gait analysis, stroke, spasticity

## Abstract

Background/Objectives: Post-stroke hemiparetic gait often presents with asymmetric patterns to compensate for stability deficits. This study examines gait differences in chronic stroke patients with spastic hemiparesis based on initial foot contact type—forefoot versus rearfoot. Methods: Thirty-four independently walking spastic hemiparetic patients were retrospectively analyzed. Using 3D gait analysis, patients were categorized by initial contact type. Spatiotemporal descriptors, joint kinematics, kinetics, and EMG patterns were compared across groups. Results: Patients with rearfoot initial contact (G1) showed higher cadence, longer single-limb support time and shorter stride times than those with forefoot contact (G0). G1 patients also demonstrated greater knee flexion during stance, enhancing stability. Additionally, G1 patients with abnormal lateral gastrocnemius activation in the swing phase showed increased ankle power at the end of the stance phase. Conclusions: In post-stroke spastic hemiparetic patients, the type of initial foot contact—forefoot or rearfoot—appears to influence gait characteristics, with rearfoot contact associated with a trend toward improved gait parameters, such as increased cadence and longer single-limb support.

## 1. Introduction

The Global Stroke Factsheet discloses a 50% increase in the lifetime risk of experiencing a stroke over the past 17 years [1]. Presently, it is estimated that one in four individuals will encounter a stroke during their lifetime, and it continues to be the third-leading cause of disability [1].

In the acute phase post stroke, 50% of patients are unable to walk, 12% can walk but need support and 37% can walk independently. After 3 months of rehabilitation, 18% of patients are still unable to walk and 11% can walk but need assistance [2]. Typically, hemiparetic stroke patients have an asymmetric gait with a pattern that tries to achieve the stability of the pelvis, with decreased stance and increased swing phase on the paretic side, decreased step length on the non-affected side and increased double-support time [3]. Along with the stride and step parameters, gait variability is also significant due to its association with clinical outcomes, including the likelihood of falls [4].

Walking velocity in chronic stroke patients has been reported to range from 0.10 m/s [5] to 0.76 m/s [6]. In fact, the hemiparetic gait pattern is characterized by an increased stride time, a reduced cadence [2,7,8] and a reduced walking velocity [9,10].

Gait analysis, as a procedure, is essential for the quantitative assessment of gait deviations, providing functional diagnosis that helps guide clinical decision-making [3]. Rather than being a singular tool or device, it encompasses a structured process that integrates observation, measurement, and interpretation to assess locomotion comprehensively. This systematic approach involves evaluating various phases of gait, including spatial and temporal parameters, joint kinematics, kinetics, and muscle activity. In the clinical field, optoelectronic 3D motion capture systems and force plates are widely recognized as the gold-standard tools for assessing gait patterns in a laboratory setting, owing to their reliability and measurement precision [11].

In normal gait, the stance phase initiates with heel contact and the talocrural joint is in slightly plantar flexed position [10]. During stance, up to 10 degrees of ankle dorsiflexion occurs as the tibia moves forward over the foot in contact with the ground. After heel-off, the ankle starts to plantarflex, reaching a maximum of 15 to 20 degrees of plantar flexion just after toe-off, and the ankle plantar flexors are the primary contributors to the development of internal ankle power for propulsion [3]. During the swing phase, the ankle is once again dorsiflexed in order to avoid contact with the ground [10].

The initial contact with the forefoot, rather than the rearfoot, is commonly observed in various types of spastic hemiplegia. According to the gait classification proposed by Winters et al. [12], four distinct patterns based on sagittal plane kinematics are identified. In Type 1 hemiplegia, a ‘drop foot’ is prominent, particularly during the swing phase of gait, resulting from the inability to selectively control ankle dorsiflexors. Notably, there is no calf contracture, allowing for relatively normal ankle dorsiflexion during the stance phase. Type 2 hemiplegia is the most prevalent in clinical practice, featuring true equinus during the stance phase due to the spasticity or contracture of the gastrocnemius-soleus muscles. An overactive plantar flexion–knee extension couple is observed, leading to the possibility of the knee adopting an extended or hyperextended position [13]. Type 3 hemiplegia is distinguished by gastrocnemius-soleus spasticity or contracture, impaired ankle dorsiflexion during the swing phase, and a ‘stiff knee gait’ characterized by hamstring/quadricep co-contraction. In Type 4 hemiplegia, there is more pronounced proximal involvement. In the sagittal plane, equinus, a flexed stiff knee, a flexed hip, and an anterior pelvic tilt are evident. In the coronal plane, hip adduction is observed, and in the transverse plane, there is internal rotation.

Recently, the identification of optimal post-stroke hemiplegic gait patterns has been enhanced by utilizing full joint-level kinematic features to form gait groups through clustering algorithms, rather than relying on traditional clinical evaluations. In one study, six distinct gait groups were identified [14]. Since these gait groups and their features can interfere with rehabilitation strategies, it is crucial to align these clusters with clinical evaluations, to define appropriate treatment plans for each patient based on their specific gait group.

Equinovarus foot deformity is commonly observed after a stroke and induces gait abnormalities. The ankle assumes a plantar flexed posture and typically presents with an inverted varus deformity [15,16]. Patients have initial contact with the forefoot, bearing weight on the anterior and lateral borders of the foot. Indeed, in the mid-stance phase, the deficit in dorsiflexion impedes the forward movement of the tibia over the foot, resulting in knee hyperextension. Kinetic studies have revealed heightened lateral plantar support, irregular force transfer from the hindfoot to the forefoot, inadequate rollover, and diminished or absent push-off during the terminal stance.

In the swing phase, equinovarus foot deformity generates limb clearance problems, resulting in observed counterbalance maneuvers such as hip hiking, trunk movement, circumduction, and contralateral vaulting. The explanation for this deformity is associated with the overactivation of ankle plantar flexors during both the swing and stance phases and weakness in ankle dorsiflexors [17]. Gait analysis can identify stiffness at the ankle due to the overactivity of the plantar flexor muscles, and spatial and temporal descriptors correlate with variations in joint powers and moments of forces [15,17]. In some adult patients with overactivity, botulinum toxin injections to the ankle plantar flexor muscles can positively influence walking velocity [3]. This analysis assists in treatment planning, including the consideration of botulinum toxin injections for cases with overactive plantar flexors [18], or surgical interventions when there is a restricted range of motion without muscle overactivity.

The primary objective of the present study is to evaluate differences in spatiotemporal descriptors, joint kinematics, kinetics, and electromyographic activation, based on whether initial contact during the stance phase occurs with the forefoot or the rearfoot, in spastic hemiparetic post-stroke patients.

## 2. Materials and Methods

Patients after a stroke capable of walking were retrospectively analyzed. A total of 116 patients were screened; they were among those sequentially referred between 2013 and 2016 to the gait analysis laboratory of the Centro de Medicina de Reabilitação de Alcoitão, Lisbon. The inclusion criteria were a post-stroke time of at least 3 months, no concurrent pathologies affecting the CNS or the neuromuscular system, the ability to walk at least 10 m without aids or assistance, and no major musculoskeletal injuries or surgeries that could impact ambulatory ability. Patients using an external walking device, an orthosis, or support by a third person were excluded. Thirty-four individuals with hemiparesis after stroke were then included in the study. The retrospective analysis of the patient files was authorized by the Ethical Committee of the Centro de Medicina de Reabilitação (project identification number 14_2020) on 5 June 2020.

The protocol consisted of a gait analysis approach using six infrared cameras (Vicon T-series T10, Vicon Motion Systems Ltd., Oxford, UK), two digital video cameras (Basler piA1000-48gc, Basler AG, Ahrensburg, Germany), four force platforms (AMTI OR6-7-2000, Advanced Mechanical Technology, Inc., Watertown, MA, USA), and a 16-channel electromyography system (Cometa Mini Wave Infinity, Cometa Systems, Bareggio, Italy).

Participants walked barefoot along a level, 10 m walkway at a self-selected, comfortable speed. The walkway included embedded force platforms to measure ground reaction forces. Reflective markers were attached to specific anatomical landmarks according to the Plug-in Gait model (Vicon Motion Systems Ltd., Oxford, UK). Prior to the experimental trials, participants performed a series of familiarization walks to ensure consistency in their gait pattern. For each participant, a minimum of five valid walking trials were collected. A valid trial was defined as one in which the participant walked smoothly across the walkway without hesitation.

The recorded marker trajectories and platform data allowed for the computation of joint kinematics and kinetics, which were normalized to anthropometric parameters such as body height, leg length and body mass. Several outcomes were extracted from patients’ gait analysis records.

Spatiotemporal descriptors, including step length, cadence, and walking speed, were automatically calculated by the system. Static kinematics parameters were calculated as averages of the analyzed period. Kinematics descriptors were determined manually by identifying the peak values of the curves. The type of initial contact of the paretic side during stance (forefoot vs. rearfoot) was subjectively analyzed on the video samples.

Statistical analysis was performed using IBM SPSS Statistics, Version 27.0 (IBM Corp., Armonk, NY, USA). The dependent variables included spatiotemporal descriptors (such as step length, stride time, cadence, walking speed), joint kinematics and kinetics of lower extremities. Differences between groups were assessed using the Mann–Whitney test for non-normally distributed variables or the independent-samples *t*-test for normally distributed variables. A probability of type 1 error *p* = 0.05 was adopted as the critical threshold for determining statistical significance.

## 3. Results

A total of 116 patients were screened, with the inclusion of all the stroke patients admitted from 2013 to 2016. Of those, 15 were excluded for walking with the aid of an external individual and 67 were excluded for walking with an orthosis or walking device, resulting in 34 eligible subjects.

### 3.1. Characterization of the Population

The mean age of the sample was 49.7 years (standard deviation (SD) ± 13.4), with 11 females and 23 males. Among the participants, 21 had right hemiparesis, and 13 had left hemiparesis. According to video samples, 17 had initial contact with forefoot and 17 contact the ground with rearfoot.

### 3.2. Spatiotemporal Descriptors

Table 1 presents the values of the spatiotemporal parameters by groups according to initial contact in the stance phase with forefoot (G0) or the rearfoot (G1).

G1 patients demonstrated a higher cadence (83.87 ± 14.85 steps/min) compared to G0 participants (72.87 ± 10.50 steps/min, *p* = 0.019). G1 also exhibited longer single-limb support times on both the paretic and non-paretic sides (*p* = 0.041) and shorter stride times on the non-paretic side (*p* = 0.017). Additionally, G1 participants showed a trend toward higher walking speed compared to G0 (0.55 m/s vs. 0.47 m/s), though this did not reach statistical significance (*p* > 0.05).

### 3.3. Joint Kinematic and Kinetics of Lower Extremities

Table 2, Table A1 and Table A2 (Appendix A) present the values for the range of motion and the maximum values of the lower-limb kinematics and kinetics during the gait cycle.

G1 participants achieved greater knee flexion during single-limb support (*p* = 0.04) and exhibited reduced plantar flexion at initial contact compared to G0 (*p* = 0.003). G1 also showed significantly greater ankle dorsiflexion during the swing phase (*p* < 0.001), suggesting improved limb clearance.

In the stance phase, typically the heel contacts the ground with the talocrural joint in a slightly plantar-flexed position. However, individuals with initial forefoot contact may experience an increased plantarflexion position. For example, during single-limb support in the stance phase, individuals with an equinus deformity may show reduced knee flexion or even knee hyperextension, as the dorsiflexion deficit hinders the forward progression of the tibia over the foot.

### 3.4. Dynamic Telemetric Electromyography Records

Table 3 presents the patients with abnormal electromyography of medial or lateral heads of the gastrocnemius muscle or soleus muscle activation during the swing phase and compares the spatial temporal descriptors, gait kinematics and kinetics of the patients with initial contact with rearfoot versus forefoot during the stance phase.

Patients with an anomalous muscle activation of EMG medial or lateral heads of gastrocnemius in G1 revealed higher cadence compared with patients in G0. Furthermore, the patients with an abnormal lateral head of gastrocnemius muscle activation whose initial contact was with the rearfoot increased ankle maximum power during the stance phase, compared with patients with the same anomalous activations whose initial contact was with the forefoot.

## 4. Discussion

Our study focused on the gait analysis of chronic post-stroke spastic hemiparetic patients with respect to the initial contact of the affected foot, distinguishing between the rearfoot (G1) and forefoot (G0). The results highlight notable distinctions in spatiotemporal parameters, joint kinematics, and electromyographic activity, offering insights into how initial contact influences gait.

Pes equinus deformity caused by spasticity of the plantar flexors and invertors or ankle plantar flexion contracture will have an impact on both the stance and swing phases of gait. Stroke patients with this deformity tend to have initial ground contact with the lateral border of the forefoot [10]. At mid-stance, lowering the heel to the ground may produce knee hyperextension due to the impossibility of the tibia to move forward over the foot. Afterwards, an excessive forward lean of the trunk with hip flexion occurs as a strategy to maintain the forward progression of the center of mass [10].

During the swing phase, pes equinus deformity creates limb clearance problems. To compensate for this, observable strategies include hip hiking, hip circumduction, or excessive hip and knee flexion in the swinging lower extremity, as well as vaulting the contralateral stance limb. These adaptations aim to prevent the toes from dragging during the swing phase.

G1 participants demonstrated higher cadence and increased single-support time compared to those in G0. They also showed a reduced time allocation to the loading response phase, enabling a faster transition to contralateral foot-off and thereby contributing to a higher cadence. These findings suggest that rearfoot contact is associated with greater confidence in weight-bearing and improved stability. While walking speed was higher in G1, this difference did not reach statistical significance. Nevertheless, the trend aligns with the observed improvements in cadence and support time, suggesting a more stable gait pattern in G1.

Joint kinematics analysis revealed that individuals with rearfoot initial contact was associated with greater knee flexion during single-limb support, which likely contributes to enhanced limb alignment and balance. Additionally, G1 participants exhibited greater ankle dorsiflexion during the swing phase, indicating better limb clearance and reduced reliance on compensatory strategies such as hip hiking or circumduction. The reduced plantar flexion at initial contact in G1 may reflect improved control over the affected limb during stance-phase initiation.

The ankle plantar flexors play a crucial role as the primary contributors to the generation of internal ankle power at late stance [3] and are responsible for the anterior projection of the body [19]. Despite the hyperactivity or shortening of the plantar flexors, hemiparetic patients typically exhibit reduced push-off power. This is related to the paresis and deficit of voluntary selective control of movement that these patients present on the affected side, as well as on the biomechanic determinants of the resultant of the ground forces at late stance. In typical gait, the second peak of the vertical ground reaction force at terminal stance corresponds to the propulsion provided by the plantar flexors [10], aiding the forward propulsion of the center of mass and increasing contralateral step length [20,21,22].

There is a wide spectrum of patients with varying degrees of plantar flexor overactivity. Participants with abnormal gastrocnemius activation in G1 showed increased ankle power during the stance phase compared to those in G0. This suggests that rearfoot contact may facilitate the more effective use of plantar flexor muscles for propulsion, despite the hyperactivity often observed in spastic hemiparesis. Higher cadence in G1 further supports the notion of improved gait dynamics associated with rearfoot contact.

Initial contact type significantly influences gait parameters in post-stroke spastic hemiparetic patients. Rearfoot contact was associated with higher cadence, longer single-limb support time, and improved knee flexion, suggesting a more stable and efficient gait pattern. In spastic patients, integrating gait analysis with clinical evaluation, including the assessment of equinus deformity, is essential for a better understanding of initial contact patterns. A comprehensive understanding of how these patterns affect gait mechanics can contribute to refining treatment approaches, such as botulinum toxin injections, the use of orthotic devices, or surgical interventions.

## 5. Conclusions

Gait asymmetry in stroke patients is an adaptation to the neurologic deficits aiming to provide stability. Under normal conditions, the gait cycle begins with initial rearfoot contact on both sides. However, this is not consistently observed in post-stroke hemiparetic patients.

This study compared gait parameters in spastic stroke patients with initial rearfoot contact versus forefoot contact. The initial contact with the forefoot is commonly observed in various types of spastic hemiplegia. Forefoot initial contact, commonly seen in various types of spastic hemiplegia, was associated with lower gait efficiency in our study, as reflected by slower walking speeds (0.47 m/s compared to 0.55 m/s), although this difference was not statistically significant in our sample. Rearfoot initial contact was associated with increased cadence, longer single-support time, and shorter stride times. Additionally, patients with abnormal gastrocnemius activation during the swing phase, who maintained rearfoot contact, demonstrated increased cadence and higher ankle power at the end of the stance phase.

## Figures and Tables

**Table 1 neurolint-17-00010-t001:** Spatial temporal descriptors for the two groups, according to initial contact in stance phase (N = 34).

Variables	G0 (n = 17)		G1 (n = 17)		*p* Value
Cadence (mean ± SD steps/min)	72.87 ± 10.50		83.87 ± 14.85		**Cadence: *p* = 0.019 (#)**
	G0 (n = 17)		G1 (n = 17)		*p* value
	PS	NPS	PS	NPS	
Double Support (median ± interquartile range s)	29.50 ± 39.72	35.60 ± 42.26	29.20 ± 16.80	31.60 ± 18.50	Double Support PS: *p* = 0.812 (*) Double Support NPS: *p* = 0.586 (*)
Single Support (median ± interquartile range s)	19.00 ± 24.69	28.90 ± 38.52	25.90 ± 13.20	38.30 ± 12.10	**Single Support PS: *p* = 0.041 (*)** **Single Support NPS: *p* = 0.041 (*)**
Foot-Off (median ± interquartile range %)	61.20 ± 8.8	75.50 ± 6.1	60.70 ± 8.1	71.50 ± 9.9	Foot-Off PS: *p* = 0.245 (*) Foot-Off NPS: *p* = 0.106 (*)
Opposite Foot Contact (median ± interquartile range %)	42.20 ± 3.0	57.60 ± 6.20	43.90 ± 3.8	56.20 ± 7.70 (&)	Opposite Foot Contact PS: *p* = 0.946 (*) Opposite Foot Contact NPS: *p* = 0.657 (*)
Opposite Foot-Off (median ± interquartile range %)	16.90 ± 9.15	21.40 ± 7.0	14.20 ± 8.90	18.20 ± 8.4	Opposite Foot-Off PS: *p* = 0.160 (*) **Opposite Foot-Off NPS: *p* = 0.034 (*)**
Limp Index (median ± interquartile range)	0.83 ± 0.14	1.28 ± 0.22	0.87 ± 0.12	1.17 ± 0.13	Limp Index PS: *p* = 0.610 (*) Limp Index NPS: *p* = 0.339 (*)
Step Time (median ± interquartile range s)	0.92 ± 0.24	0.63 ± 0.16	0.87 ± 0.29	0.59 ± 0.11	Step Time PS: *p* = 0.099 (*) **Step Time NPS: *p* = 0.024 (*)**
Stride Time	1.63 ± 0.42 (median ± interquartile range s)	1.69 ± 0.25 (mean ± SD s)	1.53 ± 0.36 (median ± interquartile range s)	1.46 ± 0.28 (mean ± SD s)	Stride Time PS: *p* = 0.057 (*) **Stride Time NPS: *p* = 0.017 (#)**
Step Length	0.36 ± 0.26 (median ± interquartile range m)	0.36 ± 0.12 (mean ± SD m) (&)	0.43 ± 0.22 (median ± interquartile range m)	0.36 ± 0.14 (mean ± SD m)	Step Length PS: *p* = 0.363 (*) Step Length NPS: *p* = 0.990 (#)
Step Width	0.26 ± 0.07 (mean ± SD m)	0.27 ± 0.090 (median ± interquartile range m)	0.22 ± 0.08 (mean ± SD m) (&)	0.20 ± 0.11 (median ± interquartile range m)	Step Width PS: *p* = 0.159 (#) Step Width NPS: *p* = 0.062 (*)
Stride Length	0.74 ± 0.43 (median ± interquartile range m)	0.67 ± 0.25 (mean ± SD m)	0.72 ± 0.49 (median ± interquartile range m)	0.78 ± 0.21 (mean ± SD m)	Stride Length PS: *p* = 0.786 Stride Length NPS: *p* = 0.204 (#)
Walking Speed (mean ± SD m/s)	0.47 ± 0.18	0.45 ± 0.17	0.55 ± 0.23	0.55 ± 0.20	Walking Speed PS: *p* = 0.226 (#) Walking Speed NPS: *p* = 0.104 (#)

Legend: G0: Initial contact of paretic side with forefoot; G1: Initial contact of paretic side with rearfoot; PS: Paretic Side; NPS: Non-Paretic Side; statistically significant correlation: *p* < 0.05; ±SD: Standard deviation; s: seconds; m: meters. Differences between the two groups G0 and G1, according to paretic and non-paretic limb were analyzed using *t*-test (#) and Mann–Whitney test (*). Variables with significant differences are in bold. (&) n = 16 patients. Data are presented as means (standard deviation) or median values (range). *p* values < 0.05 were considered statistically significant.

**Table 2 neurolint-17-00010-t002:** Values for the gait kinematics of the knee and ankle angle (N = 34).

Variable	G0 (n = 17)	G1 (n = 17)	*p* Value
Knee flexion swing-phase maximum angle (mean ± SD degrees)	31.30 ± 12.77	32.26 ± 14.75	*p* = 0.841
Knee angle at initial contact/stance phase (mean ± SD degrees)	8.30 ± 7.93	8.03 ± 9.16	*p* = 0.929
Knee extension stance-phase maximum angle (mean ± SD degrees)	−6.83 ± 10.59	−2.82 ± 9.85	*p =* 0.262
**Knee flexion single-limb support/stance-phase maximum angle (mean ± SD degrees)**	−1.29 ± 12.97	8.48 ± 13.85	** *p =* ** **0.04**
**Ankle plantarflexion swing-phase maximum angle (mean ± SD degrees)**	2.76 ± 4.20	9.99 ± 4.21	** *p* ** **= 0.000**
**Ankle at initial contact/stance-phase angle (mean ± SD degrees)**	−3.67 ± 5.38	2.00 ± 4.96	** *p* ** **= 0.003**

Legend: G0: Initial contact of paretic side with forefoot; G1: Initial contact of paretic side with rearfoot. Differences between the two groups G0 and G1 were analyzed using *t*-test. Variables with significant differences are in bold. Data are presented as means ± (SD) standard deviation. *p* values < 0.05 were considered statistically significant.

**Table 3 neurolint-17-00010-t003:** Patients with abnormal electromyographic muscle activation during the swing phase.

	GMM EMG Activation Swing Phase			GML EMG Activation Swing Phase			Solear EMG Activation Swing Phase		
	G0 (n = 4)	G1 (n = 8)		G0 (n = 3)	G1 (n = 7)		G0 (n = 6)	G1 (n = 10)	
	Mean	Mean	*p* Value	Mean	Mean	*p* Value	Mean	Mean	*p* Value
Walking speed paretic side (mean ± SD m/s)	0.55 ± 0.2	0.66 ± 0.3	0.457 (1)	0.51 ± 0.2	0.73 ± 0.2	0.175 (1)	0.49 ± 0.2	0.65 ± 0.2	0.142 (1)
Walking speed non-paretic (mean ± SD m/s)	0.55 ± 0.2	0.65 ± 0.2	0.451 (1)	0.51 ± 0.2	0.72 ± 0.2	0.132 (1)	0.48 ± 0.16	0.65 ± 0.2	0.102 (1)
Cadence (mean ± SD steps/min)	73.3 ± 3.3	89.5 ± 11.4	0.005 (1)	73.9 ± 3.8	95.0 ± 12.37	0.023 (1)	72.3 ± 7.3	90.34 ± 12.7	0.007 (1)
First minimum peak of the anterior–posterior GRF (% body weight)	(−0.64 ± 0.4)	(−0.5 ± 0.7)	0.744 (1)	(−0.51 ± 0.4)	(−0.99 ± 0.9)	0.409 (1)	(−0.65 ± 0.4)	(−0.76 ± 0.9)	0.785 (1)
Second maximum peak of the anterior–posterior GRF (% body weight)	0.65 ± 0.4	0.6 ± 0.6	0.865 (2)	0.55 ± 0.4	0.64 ± 0.6	0.732 (2)	0.53 ± 0.4	0.47 ± 0.6	0.844 (1)
First maximum peak of the vertical GRF (% body weight)	10.67 ± 0.9	10.78 ± 1.2	0.871 (1)	10.96 ± 0.8	10.94 ± 1.3	0.984 (1)	10.4 ± 0.9	10.74 ± 1.1	0.525 (1)
Second maximum peak of the vertical GRF (% body weight)	9.95 ± 0.6	9.91 ± 0.4	0.881 (1)	10.23 ± 0.4	9.84 ± 0.4	0.231 (1)	9.86 ± 0.5	9.87 ± 0.4	0.979 (1)
Ankle maximum power in stance phase (median ± interquartile range W/kg)	0.66 ± 0.8	1.16 ± 1.0	0.174 (2)	0.26 ± 0.2	1.25 ± 1.0	0.05 (2)	0.51 ± 0.7	1.04 ± 0.9	0.083 (2)
Ankle angle at foot-off (degrees)	(−3.8 ± 5)	2.85 ± 9.1	0.636 (1)	(−2.56 ± 5.4)	1.55 ± 8.9	0.210 (2)	(−1.28 ± 5.6)	3.24 ± 8.2	0.104 (2)
Ankle angle during the stance phase at initial ground contact(degrees)	(−7.32 ± 5)	3 ± 4.7	0.828 (1)	(−6.35 ± 5.7)	2.81 ± 4.9	0.031 (1)	(−5.54 ± 4.8)	2.31 ± 4.8	0.007 (1)
Maximum ankle flexion angle during the swing phase (degrees)	(−0.59 ± 5.0)	10.01 ± 3.9	0.002 (1)	(−0.04 ± 5.0)	9.02 ± 3.3	0.013 (1)	1.47 ± 5.1	9.66 ± 3.7	0.002 (1)

Legend: G0: Initial contact of paretic side with forefoot; G1: Initial contact of paretic side with rearfoot; EMG: electromyography muscle; GMM: medial head of the gastrocnemius muscle; GML: lateral head of the gastrocnemius muscle; Soleus: soleus muscle; m: meters; s: seconds; GRF: Ground reaction force. The *p* value was derived using the *t*-test (1) and Mann–Whitney Test (2). *p* values < 0.05 were considered statistically significant.

## Data Availability

The data underlying this study is accessible upon request from the corresponding author.

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
