# Peer review of "Initial Contact with Forefoot or Rearfoot in Spastic Patients After Stroke—Three-Dimensional Gait Analysis"

_2035-8377, 2025, doi:10.3390/neurolint17010010_

Round 1
Reviewer 1 Report
Comments and Suggestions for Authors
The authors sought to investigate the impact of the type of initial foot contact—forefoot versus rearfoot—on gait characteristics in post-stroke spastic hemiparetic patients. Using 3D gait analysis, the study evaluates spatiotemporal, kinematic, and kinetic parameters as well as electromyographic (EMG) patterns. It concludes that rearfoot contact is associated with improved gait parameters, such as increased cadence and longer single-limb support time, compared to forefoot contact. The manuscript is well presented, and the analysis is sound. However, the authors should address the following points that would improve the overall quality and clarity of this study:
1. Incomplete methodology: lacking experimental protocol and data reduction sections (see below)
2. Some sentences, particularly in the results and discussion sections, could be more concise or clearer. For instance, “Patients of G1 had higher cadence and single support”. What does this mean? G1 participants elicited or demonstrated higher cadence and increased single support time compared to those in G0.
3. See more specific comments below
Abstract: you listed the dependent variables twice; use general terms in the first instance, such as ambulatory parameters, etc.
Intro: Gait analysis is considered a procedure as it involves a systematic process of evaluating how someone walks or runs; essentially, it's a method of assessment rather than a single tool itself.
The purpose of the study needs some rewording. You can also add your original hypothesis. Start with: “The primary objective of the present study is……………..based on whether initial contact during the stance phase occurs with the forefoot or the rearfoot”
Methods: Your inclusion criteria should specify major musculoskeletal injuries or surgeries that could impact the participants' ambulatory ability.
When reporting devices or equipment in scientific papers, you should provide detailed information about the specific model, manufacturer and location (city, state/country) to ensure reproducibility of your study, typically including the full product name, model number and company name in parentheses following the irst mention of the device. Same for software (SPSS, motion analysis software, etc).
Experimental protocol and data reduction is incomplete. You should list all the procedural methods in the design and implementation of the experiment. Missing step-by-step information about subjects’ procedures; walking area (ground or treadmill), speed (self-paced or predetermined), number of trials, number of steps/strides, walking distance, etc.
Statistical analysis needs to be more detailed. You can list the dependent variable here, or this can be a continuation of the data reduction section that is missing.
Results: Needs to be more detailed. use legends for all abbreviated parameters or groups. There are some parameters that you haven’t mentioned (TornFapoAngChegadaSolo and others).
Discussion: While statistical methods are mentioned, it’s not always clear how results of borderline significance contribute to conclusions.
Author Response
Reviewer1
The authors sought to investigate the impact of the type of initial foot contact—forefoot versus rearfoot—on gait characteristics in post-stroke spastic hemiparetic patients. Using 3D gait analysis, the study evaluates spatiotemporal, kinematic, and kinetic parameters as well as electromyographic (EMG) patterns. It concludes that rearfoot contact is associated with improved gait parameters, such as increased cadence and longer single-limb support time, compared to forefoot contact. The manuscript is well presented, and the analysis is sound. However, the authors should address the following points that would improve the overall quality and clarity of this study:
- Incomplete methodology: lacking experimental protocol and data reduction sections (see below)
- Some sentences, particularly in the results and discussion sections, could be more concise or clearer. For instance, “Patients of G1 had higher cadence and single support”. What does this mean? G1 participants elicited or demonstrated higher cadence and increased single support time compared to those in G0.
- Answer: Thank you for pointing out the need for clearer phrasing in some sentences within the results and discussion sections. We have revised the text for greater precision and readability.
- See more specific comments below
Abstract: you listed the dependent variables twice; use general terms in the first instance, such as ambulatory parameters, etc.
- Answer: Thank you for your valuable feedback. In the revised version of the abstract, we have used remove in the first instance to avoid repetition and ensure clarity.
Intro: Gait analysis is considered a procedure as it involves a systematic process of evaluating how someone walks or runs; essentially, it's a method of assessment rather than a single tool itself.
- Answer: Thank you for your insightful feedback regarding the characterization of gait analysis. In response, we have clarified that gait analysis is indeed a procedural assessment rather than a singular tool or device. This revision highlights the procedural and systematic nature of gait analysis while underscoring its role in clinical evaluation and decision-making. We believe this modification aligns with your suggestion and improves the clarity of the introduction.
The purpose of the study needs some rewording. You can also add your original hypothesis. Start with: “The primary objective of the present study is……………..based on whether initial contact during the stance phase occurs with the forefoot or the rearfoot”
- Answer: Thank you for your constructive feedback. In response to your suggestion, we have reworded the purpose of the study for improved clarity. We also included our original hypothesis as requested. Below is the revised version:
"The primary objective of the present study is to evaluate differences in spatiotemporal descriptors, joint kinematics, kinetics, and electromyographic activation based on whether initial contact during the stance phase occurs with the forefoot or rearfoot in spastic hemiparetic post-stroke patients. "
Methods: Your inclusion criteria should specify major musculoskeletal injuries or surgeries that could impact the participants' ambulatory ability.
- Answer: Thank you for your valuable suggestion. We have updated the inclusion criteria to specify that participants with major musculoskeletal injuries or surgeries impacting ambulatory ability were excluded. This additional detail ensures clarity and reinforces the robustness of our participant selection process.
When reporting devices or equipment in scientific papers, you should provide detailed information about the specific model, manufacturer and location (city, state/country) to ensure reproducibility of your study, typically including the full product name, model number and company name in parentheses following the first mention of the device. Same for software (SPSS, motion analysis software, etc).
- Answer: Thank you for highlighting the importance of providing detailed information about the equipment and software used in the study to ensure reproducibility. We have updated the manuscript to include the full product names, model numbers, manufacturers, and locations for all devices and software. The revised text now specifies:
- Infrared cameras: Vicon T-series T10 (Vicon Motion Systems Ltd., Oxford, UK)
- Digital video cameras: Basler piA1000-48gc (Basler AG, Ahrensburg, Germany)
- Force platforms: AMTI OR6-7-2000 (Advanced Mechanical Technology, Inc., Watertown, MA, USA)
- Electromyography system: Cometa Mini Wave Infinity (Cometa Systems, Bareggio, Italy)
- Statistical software: IBM SPSS Statistics for Windows, Version 27.0 (IBM Corp., Armonk, NY, USA)
These details have been incorporated into the manuscript following the first mention of each device or software. We hope this ensures clarity and enhances the reproducibility of the study.
Experimental protocol and data reduction is incomplete. You should list all the procedural methods in the design and implementation of the experiment. Missing step-by-step information about subjects’ procedures; walking area (ground or treadmill), speed (self-paced or predetermined), number of trials, number of steps/strides, walking distance, etc.
- Answer: Thank you for your suggestion to provide more comprehensive details about the experimental protocol and data reduction process. We have revised the text to include step-by-step information about subject procedures, including the walking area, speed (self-selected, comfortable), number of trials (a minimum of five valid trials per participant), and data collection. We also clarified how data were processed, including methods for spatiotemporal descriptor calculations, static and dynamic kinematics, and video analysis. We hope these additions improve the clarity and reproducibility of the methodology.
Statistical analysis needs to be more detailed. You can list the dependent variable here, or this can be a continuation of the data reduction section that is missing.
- Answer: Thank you for your comment regarding the need for more detail in the statistical analysis section. We have revised the text to include the dependent variables such as spatiotemporal descriptors, kinematics and kinetics. Additionally, we clarified the statistical methods, including the criteria for selecting between the Mann-Whitney and independent-samples T-tests. These changes aim to provide a more comprehensive explanation of the statistical approach and its relationship to the data reduction process.
Results: Needs to be more detailed. use legends for all abbreviated parameters or groups. There are some parameters that you haven’t mentioned (TornFapoAngChegadaSolo and others).
- Answer: Thank you for your suggestion to provide more detailed explanations and include legends for all abbreviated parameters or groups. We have updated the manuscript to clarify the meaning of all parameters and ensure comprehensive coverage of the gait variables. Specifically, the parameter "TornFapoAngChegadaSolo" has been detailed, along with any additional parameters that were previously not explicitly mentioned.
Discussion: While statistical methods are mentioned, it’s not always clear how results of borderline significance contribute to conclusions.
- Answer: We appreciate your feedback regarding the clarity of how results of borderline significance contribute to our conclusions. Specifically:
Walking Speed Trend: Although the increase in walking speed for G1 did not reach statistical significance, we highlighted how this aligns with other statistically significant improvements (e.g., higher cadence and longer single-limb support time).

Reviewer 2 Report
Comments and Suggestions for Authors
The methodology employed by the authors is rigorous and well-designed. They utilize state-of-the-art techniques and tools, ensuring the validity and reliability of their results. The inclusion of experimental evidence strengthens the credibility of their claims and supports their conclusions. While there are some minor grammatical errors throughout the paper, they do not significantly hinder the readability or comprehension.
Some minor issues currently exist:
1 The data analysis presented in Tables 1 to 3 is somewhat inadequate, particularly for Table 3, which necessitates further quantitative analysis.
2 A majority of the referenced literature appears outdated, especially within the Introduction section; an ample supply of up-to-date references is required to substantiate the significance of this research study.
Author Response
Reviewer 2
The methodology employed by the authors is rigorous and well-designed. They utilize state-of-the-art techniques and tools, ensuring the validity and reliability of their results. The inclusion of experimental evidence strengthens the credibility of their claims and supports their conclusions. While there are some minor grammatical errors throughout the paper, they do not significantly hinder the readability or comprehension.
Some minor issues currently exist:
- The data analysis presented in Tables 1 to 3 is somewhat inadequate, particularly for Table 3, which necessitates further quantitative analysis.
Answer: Thank you for highlighting the need for additional quantitative analysis in Table 3. We have reformatted Table 3 for clarity to ensure the data supports the narrative in the text. The Discussion section now provides a clearer explanation of the clinical relevance of the results. We hope the revisions address your concerns and enhance the manuscript’s rigor.
- A majority of the referenced literature appears outdated, especially within the Introduction section; an ample supply of up-to-date references is required to substantiate the significance of this research study.
Answer: Thank you for highlighting the importance of incorporating up-to-date references to strengthen the Introduction section. We have reviewed the cited literature and replaced several outdated references with more recent and relevant studies that align with the current state of research. This update ensures the context and significance of our study are substantiated with the most current findings.
